# Preliminary Analysis of COVID-19 Vaccination Factors among Native and Foreign-Born Hispanic/Latine Adults Residing in South Florida, U.S.A.

**DOI:** 10.3390/ijerph192013225

**Published:** 2022-10-14

**Authors:** María Eugenia Contreras-Pérez, Janet Diaz-Martinez, Robbert J. Langwerden, Michelle M. Hospital, Staci L. Morris, Eric F. Wagner, Adriana L. Campa

**Affiliations:** 1Community-Based Research Institute, Florida International University, Miami, FL 33199, USA; 2Research Center in a Minority Institution, Florida International University, Miami, FL 33199, USA; 3School of Social Work, Florida International University, Miami, FL 33199, USA; 4Department of Biostatistics, Florida International University, Miami, FL 33199, USA; 5Caridad Center, Boynton Beach, FL 33472, USA; 6Department of Dietetics and Nutrition, Florida International University, Miami, FL 33199, USA

**Keywords:** SARS-CoV-2, pandemic, vaccination, immigrant, ethnic group

## Abstract

This study explored barriers, motivators, and trusted sources of information regarding COVID-19 vaccination among Hispanic/Latine individuals. Hispanic/Latine is a broad social construct that encompasses people from heterogeneous countries and cultures. In the U.S., foreign-born Hispanics/Latines tend to have better health outcomes than U.S.-born individuals. Thus, the study examined whether nativity is a significant factor in COVID-19 vaccine hesitancy. Binary logistic regression and linear regression analyses were employed and revealed that, regardless of nativity, Hispanic/Latine participants face similar barriers and find similar sources of information trustworthy. Controlling for age and race, vaccination rates or perceived likelihood of getting vaccinated did not differ between the two groups. The two groups significantly differed in specific motivators for vaccination: foreign-born Hispanic/Latine individuals were more motivated to get the vaccine to keep themselves, their families, and their community safe, and more often believed vaccination is needed for life to return to normal. Study results provide important insights into similarities and differences in barriers, motivators, and trusted sources of information regarding COVID-19 vaccination among native and foreign-born Hispanic/Latine individuals.

## 1. Introduction

Since the beginning of the pandemic of SARS-CoV-2, efforts have been made to reduce contagion rates and ameliorate severe illness or death. Recent research indicates that vaccination is the best long-term strategy to prevent contagion by achieving herd immunity, which will require 60% to 90% of the population to be immune [1]. Unfortunately, a variety of barriers make reaching this goal very challenging [2,3] and decreases the chances of reaching herd immunity.

One of the most salient COVID-19-related public health challenges is vaccine hesitancy [3]. A recent systematic review and meta-analysis showed that the percentage of people intending to receive a COVID-19 vaccine has consistently decreased, which shows the need to address social inequalities that may be increasing vaccine hesitancy [4]. Additionally, it has been reported that lower intention to get vaccinated is associated with identifying as female, younger age, low income, lower education level, relying heavily on social media in favor of broadcast media, low trust in science, health care providers and the government, and identifying as a racial or ethnic minority [5,6].

Understanding the factors influencing vaccine hesitancy among ethnic minorities is crucial to more efficient tailoring of vaccination campaigns for these groups [4,7]. This is particularly important in communities where most of the population identifies as racial or ethnic minority. The present study focuses on South Florida, where a vast percentage of the population identifies as Hispanic/Latine (Miami Dade-County 72% [7]; Broward County 32% [8]; Palm Beach County 24% [9]). Recent studies have sought to identify the variables that might increase vaccination hesitancy among Hispanic/Latine individuals. A recent study by Urrunaga-Pastor and colleagues of a half a million adults from 20 Latin American countries found eight out of ten individuals had intentions to get vaccinated if available. Still, eight out of ten reported also being afraid of adverse effects [10]. Overall, studies with Hispanic/Latine participants living in Latin America have found that individuals are more likely to get vaccinated when the information comes from medical experts, when the source provides general vaccine information, and when it promotes the social approval benefits of vaccination [11]. This underlines the importance of the source and content of COVID-19 vaccine information for Hispanic/Latine individuals.

Compared to U.S.-born Hispanics/Latines, immigrant Hispanics/Latines experience more social disadvantages, yet are less likely to commit a crime and present less antisocial behavior [12]; report lower alcohol use (the more enculturated, the better the outcome) [13]; live an average of 10–15% longer, despite the socio-economic disadvantages [14]; and experience higher rates of stress and health risks [15]. This phenomenon is called the immigrant paradox or the Hispanic epidemiological paradox. It indicates that although this group experiences increased hardships, foreign-born Hispanics/Latines have better health outcomes and live longer than native-born Americans. This paradox could have implications for COVID-19 vaccination intention by making Hispanic/Latine individuals more willing to engage in health protective behaviors. However, this has not been confirmed since studies have shown contradictory results regarding Hispanic/Latine acceptance of the COVID-19 vaccine; different reports have shown the same, lower, and higher acceptance when comparing them non-Hispanic/Latine counterparts [16].

Notably, Hispanic/Latines who continue speaking Spanish (bilingual or not) after immigration are at a significantly lower risk of being diagnosed with a substance use disorder [17]. Despite the barriers to accessing mental health services, they are less likely to be diagnosed with mood, anxiety, or personality disorders [18]. Unfortunately, these protective factors typically decrease with each generation since foreign-born individuals from racial and ethnic minorities are significantly less likely to meet the criteria for a mood or anxiety disorder when comparing them with U.S.-born individuals also from racial and ethnic minority groups [18]. Studies have reported that Hispanic/Latine individual’s health outcomes decline not long after migration [19], which indicates that the apparent health advantage they present decreases the longer they live in the U.S. It is relevant to consider that these advantages may not apply to undocumented Hispanic/Latine immigrants since this population consistently reports worse health outcomes than U.S. born individuals [20]. Taking this into consideration, studying health decision-making, similarities, and differences between foreign-born and U.S. born individuals is crucial to better understand how to develop effective campaigns for the Hispanic/Latine population.

Few studies have focused on barriers and motivators associated with COVID-19 vaccination among Hispanics/Latines. Furthermore, most investigations have focused on the barriers, highlighting adverse socioeconomic factors, low level of education, lack of awareness of the disease and vaccines, and religious and cultural beliefs [21]. The present study investigates the motivators, barriers, and trusted sources of information that might influence the decision to get vaccinated among Hispanic/Latine adults living in South Florida. Specifically, the current study examined putative differences in endorsement of motivators, barriers, and trusted sources of information between native-born and foreign-born Hispanics/Latines. We hypothesized that foreign-born Hispanic/Latines will identify significantly fewer barriers to vaccination, significantly more motivators for vaccination, and greater trust in sources of information such as direct medical advice.

## 2. Materials and Methods

### 2.1. Participants

In 2021 several authors of this study collaborated on two separate studies that involved collecting data regarding attitudes and beliefs about COVID-19 vaccines. Since the two studies’ survey questions were nearly identical, we combined the data for the current investigation. 

The first portion of the sample were individuals who attended one of seven COVID-19 literacy townhalls between 18 March 2021 and 19 August 2021. Please see Wagner and colleagues for more information on sample recruitment and background information [22]. The second portion of the sample originated from a FL-CEAL recruitment effort by five partnering community-serving organizations in Miami-Dade, Broward, and Palm Beach counties [23]. These organizations focus on the larger South Florida metropolitan area, with a particular focus on Hispanic/Latine, Haitian, and underserved communities. These surveys were collected in May 2021, and have been previously reported on by Langwerden and colleagues [24]. Both samples were recruited within the community and were anonymous.

After merging the two samples, we included only those participants who identified as Hispanic/Latine and those who provided a valid response to the nativity question. We also excluded anyone below the age of 18. In addition, the analyses regarding vaccination status were only done with the eligible vaccination subsample of the larger Hispanic/Latine sample, excluding all survey responses from before 5 April 2021, when not all Florida adults were eligible for COVID-19 vaccination. Demographics for our final sample are shown in Table 1.

### 2.2. Instruments and Data Preparation

The NIH Community Engagement Alliance (CEAL) is funded by the National Institute on Minority Health and Health Disparities (NIMHD) and the National Heart, Lung, and Blood Institute (NHLBI) to work closely with local minority communities; CEAL’s goal is to provide trustworthy, science-based information through active community engagement and outreach to the people hardest-hit by the COVID-19 pandemic, with the goal of building long-lasting partnerships as well as improving diversity and inclusion in our research response to COVID-19 [25]. We used data from the online REDCap survey as designed by the Florida CEAL group [24,26]. 

In the current study, we assessed vaccination status (“Have you received the COVID-19 vaccine? Please select yes if you received just one of the two vaccinations or if you received both”; 0—no, 1—yes), vaccination intention (“How likely are you to get an approved COVID-19 vaccine when it becomes available?”; 1—not likely at all, to 7—very likely), eight independent motivators for vaccination (“Which of the following reasons would motivate you to get (or have motivated you to get) an approved COVID-19 vaccine? Please select all that apply”; 0—unchecked, 1—checked), nine independent barriers for vaccination (“Which of the following reasons would prevent you from getting an approved COVID-19 vaccine? Please select all that apply” 0—unchecked, 1—checked), and eleven independent trusted sources of information (“How much do you trust each of these sources to provide correct information about COVID-19?”; 1—not at all, 2—a little, 3—a great deal).

For more details on the specific items and survey procedures, we refer to Langwerden and colleagues [24]. In South Florida, on average, one of every three persons speaks a language other than English in their household [7,8,9]; therefore the survey was offered in English, Spanish, and Haitian-Creole.

### 2.3. Analysis Procedures

Pearson Chi-Square Tests and Mann–Whitney U Tests were conducted to test whether statistical differences existed between the two groups (U.S. born and foreign-born) on 30 outcome variables (see Table 2). For the research questions we conducted regression analyses using Mplus Version 8.0 [27]. We regressed the 30 outcome variables onto the dichotomous nativity independent variable (born in the U.S. vs. not born in the U.S.). We applied binary logistic regression to dichotomous outcome variables: (1) vaccination status, (2) eight motivator variables, and (3) nine barrier variables. Linear regression analyses were conducted for the following outcome variables: (1) vaccination likelihood (rate on a 7-point Likert scale ranging from 1: not at all likely to 7: very likely) among unvaccinated participants only and (2) trust in eleven sources of information (rated on a 3-point Likert scale: 1: not at all, 2: a little, 3: a great deal). All outcome variables were input into separate regression models using the Maximum Likelihood Rotation (MLR) function for linear regression and Maximum Likelihood (ML) for binary logistic regression. For each analysis, age and race were included as covariates. We outputted odds ratios and odd ratio intervals in addition to *R*^2^ information.

## 3. Results

The sample size and sample composition can be found in Table 1. The results of the Pearson Chi-Square Tests indicated that participants who were born outside the U.S. reported significantly higher vaccination rates than those born in the U.S (*p* = 0.02). Participants born outside the U.S. also were statistically significantly older (*p* < 0.01), spoke a language other than English statistically significantly more often (*p* < 0.01) and were statistically significantly less likely to identify as Black or African American (*p* = 0.03). In subsequent analyses, we controlled for age and race to account for these structural differences between the groups. 

For participants eligible to receive the vaccine at the time of survey completion (*n* = 108), we regressed vaccination status onto nativity, holding age and race constant. The results, presented in Table 2, indicated that there was no statistically significant relationship between nativity and vaccination status (*p* = 0.31). Moreover, among the unvaccinated participants (*n* = 39), there was no relationship between nativity and vaccination likelihood (*p* = 0.78).

Holding age and race constant, the motivator “I want to keep my family safe” was 4.53 times more likely to be endorsed (95% CI [1.67, 12.29) by foreign-born participants than U.S.-born participants (*R*^2^ = 0.26, *p* = 0.003). Additionally, “I want to keep my community safe” was 2.47 times more likely (95% CI [1.06, 5.76) (*R*^2^ = 0.17, *p* = 0.04), “I want to keep myself safe” was 3.49 times more likely (95% CI [1.4, 8.67) (*R*^2^ = 0.21, *p* = 0.007), and “I believe life won’t go back to normal until most people get a COVID-19 vaccine” was 2.41 times more likely (95% CI [1.07, 5.45) (*R*^2^ = 0.13, *p* = 0.03) to be endorsed by foreign-born participants. We did not find any statistically significant relationship between nativity and any of the barrier or trusted sources of information variables.

## 4. Discussion

The current study compared motivators, barriers, and trusted sources of information regarding COVID-19 vaccination among a community sample of foreign-born and U.S.-born Hispanic/Latine adults living in South Florida. The study was conducted in the South Florida metropolitan area, where a wide percentage (42.6%) of the population identifies as Hispanic/Latine [8,9,10], which makes it one of the few places in the U.S that has this degree of heterogeneous Hispanic/Latine immigrant population [26]. It is essential to acknowledge that Hispanic/Latine is a broad social group that includes people from heterogeneous countries and cultures. 

We hypothesized that foreign-born Hispanic/Latines would identify significantly fewer barriers to vaccination and greater trust in sources of information such as direct medical advice. Native-born and foreign-born participants reported similar amounts of barriers and trusted sources of information, which does not support our hypothesis. However, statistically significant differences were found between the two groups in four vaccine motivators, which does align with the original hypothesis. Foreign-born participants were more likely to endorse (1) keeping themselves, (2) their families and (3) their community safe, and more often endorsed the (4) belief that life won’t go back to normal until most people are vaccinated.

Previous studies have found individuals from ethnic minorities report higher rates of vaccine hesitancy [5,6]. In the current study, foreign-born participants reported higher motivation to vaccinate even though the sources that inform their decisions seem to be the same as U.S.-born participants. The two groups reported comparable levels of trust, and consistent with earlier studies, the participants were more trusting of messages from medical experts (including their healthcare provider and the U.S. Coronavirus Task Force) than from other media [11].

Foreign-born Hispanic/Latine individuals were significantly more likely to endorse several motivators of vaccination. This result is congruent with results from previous studies that have found better health outcomes among foreign-born Hispanics/Latines [14,15,16]. This phenomenon may be driven by factors such as *familismo* and a collectivist mindset [26,28,29] among Hispanic/Latine immigrants. Notably, vaccination intention and vaccination status, after controlling for age and race, were not significantly different between U.S.-born and foreign-born participants. Foreign-born participants in the sample were significantly older. Therefore, they were eligible for vaccination first. Nonetheless, it is interesting that foreign-born participants still reported higher motivation towards vaccination for community health reasons regardless of already being vaccinated. This could be related to the process of acculturation and enculturation [11,12].

Acculturation refers to the cultural changes that emerge at a personal and group level when the individual is in sustained contact with two or more distinct cultures. It results in the gradual incorporation of the host culture’s beliefs, values, behaviors, and language [30]. Enculturation refers to how individuals adapt these variables from their heritage culture and develop a sense of belonging to their ethnic group. In the case of the sample in the current study, acculturation would be the process through which a Hispanic/Latine individual would adapt to the U.S. culture, and enculturation would entail adapting to the culture of their country of origin to retain identification with their ethnic background [30]. Acculturation, by definition, involves some loss of the ethnic culture, although both acculturation and enculturation can co-occur. Language has been studied under these factors, showing strong negative associations with acculturation and enculturation, meaning that people in direct contact with two cultures use both languages to a certain degree [31]. The implications of acculturation and enculturation must be considered when attempting to gain insight into the extent and nature of vaccination hesitancy in immigrant populations [32,33]. This could also shed light on why these individuals are more motivated to get vaccinated than their native-born counterparts. 

The U.S.-born group included significantly more individuals who identified as Hispanic/Latine and Black/African American. A recent survey regarding the COVID-19 vaccine showed that only 14% of Black Americans reported they trust in vaccine safety, and only 18% think the vaccine is effective [32]. Hispanic/Latine individuals who also identify as Black or African American are an understudied population with intersectional identities [33]; these results show how imperative it is to address the disparities experienced overall for the Hispanic/Latine Black African American communities.

### Limitations

The present study has limitations. First, nativity was a key variable but the study lacked important details such as country of origin or duration of residing in the U.S., limiting the ability to disaggregate the results and explore possible mechanisms of association between the factors studied. Another limitation of this study is the small sample size. That is why the results of this analysis must be considered preliminary. This article used a subsample of the samples analyzed in two previous studies, only including participants who identified as Hispanic/Latine and had a valid response to the nativity question, which reduced the number of participants that could be included. Finally, due to the importance of analyzing the impact of acculturation and enculturation, we suggest that studies with immigrant populations include specific measurements for these variables. 

## 5. Conclusions

We explored barriers, motivators, and trusted sources of information regarding COVID-19 vaccination among foreign-born versus U.S.-born Hispanics/Latines residing in South Florida. We found all participants, regardless of nativity, endorsed barriers in equal rates and trust in similar sources of information. After controlling for age and race, foreign-born and U.S. born participants showed similar vaccination rates. Significant differences between the two groups were found for motivators for vaccination. Foreign-born Hispanic/Latine individuals were significantly more motivated to get the vaccine to keep their family, their community, and themselves safe. They also were more likely to endorse pursuing COVID-19 vaccination because they do not believe life will return to normal until most people are vaccinated. We would like to highlight the fact that this study has three particular unique strengths: (1) the information used to build this article comes from community-based research intended to assess and elevate the community experience (as is in line with the aims of the national CEAL initiative [23], which part of the sample was collected under); (2) data collection was conducted at a crucial time for COVID-19 vaccination that allowed to gather information when vaccination was recently available for Floridian adults of all ages; and (3) data collection was conducted in multiple languages for it to adapt to the diverse population of South Florida.

This study showed that even though they have similar vaccination rates and vaccination intention, Hispanic/Latine individuals born outside the U.S. are motivated to get vaccinated compared to their native-born counterparts. this knowledge is essential to better tailor vaccination campaigns for COVID-19, including booster doses, as well as other diseases. 

## Figures and Tables

**Table 1 ijerph-19-13225-t001:** Demographic Characteristics of Survey Respondents: overall and by sample.

Characteristic	Hispanic/Latine Total *N* = 127	Sample A Hispanic/Latine Born in the U.S *n* = 75	Sample B Hispanic/Latine Not Born in the U.S *n* = 52
Age	*M* = 41.38 *SD* = 17.07 18–82	*M* = 36.11 *SD* = 15.04 18–78	*M* = 49 *SD* = 17.07 18–82
Gender			
Female	97 (77.6%)	56 (75.7%)	41 (80.4%)
Male	28 (22.4%)	18 (24.3%)	10 (19.6%)
Race White	82 (64.6%)	41 (54.7%)	41 (78.8%)
Black or African American	27 (21.3%)	21 (28%)	6 (11.5%)
Asian	7 (5.5%)	5 (6.7%)	2 (3.8%)
American Indian or Alaska Native	2 (1.6%)	0 (0%)	2 (3.8%)
Prefer not to answer	4 (3.1%)	3 (4%)	1 (1.9%)
Sexual Orientation Bisexual	5 (4.1%)	4 (5.5%)	1 (2%)
Gay	4 (3.3%)	3 (4.1%)	1 (2%)
Lesbian	2 (1.6%)	2 (2.7%)	0 (0%)
Straight	111 (81%)	64 (87.7%)	47 (95.9%)
Having health insurance			
Yes	117 (95.1%)	68 (94.4%)	49 (96.1%)
No	5 (4.1%)	3 (4.2%)	2 (3.9%)
Don’t know	1 (0.8%)	1 (1.4%)	0 (0%)
Speaking a language other than English at home No	39 (31.5%)	35 (47.9%)	4 (7.8%)
Yes	85 (68.5%)	38 (52.1%)	47 (92.2%)
Spanish	73 (57.5%)	31 (41.3%)	42 (80.8%)
Haitian Creole	4 (3.1%)	1 (1.3%)	3 (5.8%)
Portuguese	2 (1.6%)	0 (0%)	2 (3.8%)
French	3 (2.4%)	0 (0%)	3 (5.8%)
Household income before taxes Less than $15,000	9 (9%)	7 (11.7%)	2 (5%)
$15,000–$19,999	7 (7%	4 (6.7%)	3 (7.5%)
$20,000–$24,999	4 (4%)	2 (3.3%)	2 (5%)
$25,000–$34,999	6 (6%)	2 (3.3%)	4 (10%)
$35,000–$49,999	17 (17%)	8 (13.3%)	9 (22.5%)
$50,000–$74,999	19 (19%)	11 (18.3%)	8 (20%)
$75,000–$99,999	14 (14%)	10 (16.7%)	4 (10%)
$100,000 and above	24 (24%)	16 (26.7%)	8 (20%)

Results present valid percentages. The Gender variable included additional options that were not included in this table since the participants did not select them.

**Table 2 ijerph-19-13225-t002:** Chi-square and regression analyses comparing participants born in the U.S. vs. not born in the U.S.

	Hispanic/Latine Total *N* = 127		Sample A Hispanic/Latine Born in the U.S *n* = 75		Sample B Hispanic/Latine Not Born in the U.S *n* = 52		Statistical Significance between Sample A and B Chi-Square and Mann–Whitney U	Statistical Significance between Sample A and B Regression (age/race)
Vaccination status * ^a^	*n* (%)		*n* (%)		*n* (%)		*p*	*p*
Unvaccinated at time of survey completion	39 (35.8%)		29 (44.6%)		10 (23.3%)		**0.02**	0.31
Vaccination intention ^b^	*M* (*SD*)		*M* (*SD*)		*M* (*SD*)		*p*	*p*
Likelihood of getting vaccinated among unvaccinated (Likert scale average between 1 and 7)	4.64 (2.35)		4.62 (2.5)		4.69 (2.02)		0.93	0.78
Motivators for vaccination ^a^	Selected yes *n* (%)		Selected yes *n* (%)		Selected yes *n* (%)		*p*	*p*
I want to keep my family safe.	85 (66.9%)		41 (54.7%)		44 (84.6%)		**<0.001**	**0.003**
I want to keep my community safe.	72 (56.7%)		35 (46.7%)		37 (71.2%)		**0.006**	**0.04**
I want to keep myself safe.	78 (61.4%)		38 (50.7%)		40 (76.9%)		**0.003**	**0.007**
I have a chronic health problem, like asthma or diabetes.	13 (10.2%)		4 (5.3%)		9 (17.3%)		**0.03**	0.07
My doctor told me to get a COVID-19 vaccine.	7 (5.5%)		3 (4.0%)		4 (7.7%)		0.37	0.45
I don’t want to get really sick from COVID-19.	58 (45.7%)		30 (40%)		28 (53.8%)		0.12	0.42
I want to feel safe around other people.	69 (54.3%)		34 (45.3%)		35 (67.3%)		**0.02**	0.08
I believe life won’t go back to normal until most people get a COVID-19 vaccine.	59 (46.5%)		28 (37.3%)		31 (59.6%)		**0.01**	**0.03**
Barriers to vaccination ^a^	Selected yes *n* (%)		Selected yes *n* (%)		Selected yes *n* (%)		*p*	*p*
I’m allergic to vaccines.	10 (7.9%)		5 (6.7%)		5 (9.6%)		0.54	0.97
I don’t like needles.	11 (8.7%)		8 (10.7%)		3 (5.8%)		0.34	0.76
I’m not concerned about getting really sick from COVID-19.	6 (4.7%)		6 (8.0%)		0 (0%)		**0.04**	0.97
I’m concerned about side effects from the vaccine.	45 (35.4%)		31 (41.3%)		14 (26.9%)		0.1	0.08
I don’t think vaccines work very well.	6 (4.7%)		4 (5.3%)		2 (3.8%)		0.7	0.94
I don’t trust that the vaccine will be safe.	19 (15%)		14 (18.7%)		5 (9.6%)		0.16	0.47
I don’t believe the COVID-19 pandemic is as bad as some people say it is.	5 (3.9%)		4 (5.3%)		1 (1.9%)		0.33	0.19
I don’t want to pay for it.	7 (5.5%)		4 (5.3%)		3 (5.8%)		0.92	0.98
I don’t know enough about how well a COVID-19 vaccine works.	26 (20.5%)		16 (21.3%)		10 (19.2%)		0.77	0.73
Trusted sources of information ^b^	Selected yes *n* (%)		Selected yes *n* (%)		Selected yes *n* (%)		*p*	*p*
	a little	a great deal	a little	a great deal	a little	a great deal		
Your doctor or health care provider	20 (15.7%)	102 (80.3%)	12 (16%)	61 (81.3%)	8 (15.4%)	41 (78.8%)	0.87	0.52
Your faith leader	35 (27.6%)	30 (23.6%)	20 (26.7%)	20 (26.7%)	15 (28.8%)	10 (19.2%)	0.14	0.35
Your close friends and members of your family	55 (43.3%)	46 (36.2%)	31 (41.3%)	32 (42.7%)	24 (46.2%)	14 (26.9%)	0.1	0.12
People you go to work or class with or other people you know	61 (48%)	32 (25.2%)	33 (44%)	24 (32%)	28 (53.8%)	8 (15.4%)	0.19	0.07
News on the radio, TV, online, or in newspapers	65 (51.2%)	30 (23.6%)	42 (56%)	17 (22.7%)	23 (44.2%)	13 (25%)	0.97	0.99
Your contacts on social media	48 (37.8%)	8 (6.3%)	30 (40%)	6 (8.0%)	18 (34.6%)	2 (3.8%)	0.29	0.38
The U.S. government	58 (45.7%)	48 (37.8%)	33 (44%)	27 (36%)	25 (48.1%)	21 (40.4%)	0.13	0.16
The U.S. Coronavirus Task Force	34 (26.8%)	79 (62.2%)	23 (30.7%)	45 (60%)	11 (21.2%)	34 (65.4%)	0.28	0.2
Leaders in your community	56 (48.7%)	39 (33.9%)	36 (52.9%)	22 (32.4%)	20 (42.6%)	17 (36.2%)	0.91	0.48
Local Politicians	62 (53.9%)	10 (8.7%)	39 (56.5%)	5 (7.2%)	23 (50%)	5 (10.9%)	0.97	0.9
Billboards	47 (37%)	8 (6.3%)	29 (38.7%)	4 (5.3%)	18 (34.6%)	4 (7.7%)	0.94	0.83

* Participants who were not eligible for vaccination at the time they took the survey were excluded from these analyses. For these analyses the denominator was 109. Results present valid percentages. Statistically significant results are bolded; ^a^ Pearson Chi-Square Tests were conducted for vaccination status, the motivators, and barriers; ^b^ Mann–Whitney U Tests were conducted for vaccination likelihood and trusted sources of information.

## Data Availability

Not applicable.

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
