# Peer review of "Preliminary Analysis of COVID-19 Vaccination Factors among Native and Foreign-Born Hispanic/Latine Adults Residing in South Florida, U.S.A."

_ijerph, 2022, doi:10.3390/ijerph192013225_

Round 1

Reviewer 1 Report

Dear Authors

Your paper is good and comprehensive. I have read it with a great deal of interest and known many new facts about the situation with the vaccination in Florida. Thank you for that.

I hope that my remarks may help you to improve your article. If you pay due attention to them, I shall gladly recommend your text for being published in IJERPH.

Minor comments:

1. I recommend correcting the title as follows “Preliminary Analysis of COVID-19 Vaccination Factors Among Native and Foreign-Born Hispanic/Latine Adults Residing in South Florida, USA”.

2. Key words may be effectively changed. Key words are normally used to increase visibility of an article. But any search engine looks by key words and the title simultaneously. Therefore, if you wish that more people had found your publication, you may resort to a little trick. Use different key words from the words you used in the title. In addition, in your current case it is unavailing to insert “Latino; Latina; Latine”. This does not add to visibility (only a word root is being searched by engines). I would use the following list of key words: SARS-CoV-2; pandemic; immigrant; epidemiology; ethnic group. Or something like that.

3. Lines 36-38. Starting a paper with platitudes is not good. Please remove the first two sentences.

4. Line 46. “…has consistently decreased [5]”. Either expound or remove. That reference is just hanging in the air without leading to a point.

5. Line 47. “…less education level,…” Lower education level?

6. Line 65. “…Hispanic Latine…”. Hispanic / Latin?

7. Line 78. “…with white participants [17]”. With White participants? (you are referring to the ethnic group of US citizens, not the colour of skin)

8. Lines 97-100. This sentence is irrelevant in the context of your paper. Please remove it.

9. Lines 153-154. Please add to this sentence the statistic info how many people speak Spanish and Haitian-Creole in Florida, with a corresponding reference in the list of literature.

10. Line 196. I would abstain from making a separate subchapter “Figures, Tables and Schemes”. Besides, you do not have figures, nor schematics.

11. Line 213. The Constructivist paradigm is not much used among researchers of ethnicity or race even in the US, to say nothing about the rest of the world. I recommend changing “construct” to “group”.

12. Lines 214-218. Misleading sentences! Please remove them.

13. Line 237. “…antisocial behavior [13]…” How is the antisocial behaviour related to the vaccination propensity you are researching? And what antisocial behaviour do you mean? Either explain or remove.

14. Line 261-262. “Regarding … culture”. A misleading sentence. Please remove.

15. Lines 265-267. Please relocate this statement to a separate “Limitations” subchapter that should be created below. There you may specify that you did not measure acculturation or enculturation variables in this study but they are important to include in further research.

16. Lines 272-285. From this paragraph please only retain the sentences “The U.S.-born group included significantly more individuals who identified as Hispanic/Latine and Black/African American simultaneously. A recent survey regarding the COVID-19 vaccination showed that only 14% of Black Americans reported they trust in vaccine safety, and only 18% think the vaccine is effective [35]. Hispanic/Latine individuals who also identify as Black or African American are an understudied population with intersectional identities [39].” This makes the paragraph readable and meaningful. The rest is irrelevant political-correct verbosity that is not needed in a scientific paper.

17. After that, please make a separate subchapter “Limitations”.

18. Line 286. Please remove “as”.

19. Line 286. “but lacked”. Did you mean “but the study lacked”?

20. Line 288. “to further disaggregate”. A split infinitive! Please correct.

21. Line 289. “…which is why…”. Did you mean “…is the small sample size. That is why…”?

22. Lines 290-297. I would suggest relocating your strengths to “Conclusions”.

23. Lines 307-308. “…they don’t believe life will return to normal until most people are vaccinated.” Like anyone else, I dare say. So please reformulate it in a more consistent way. The final sentence of your paper must be a summarizing one. Please do not finalize the article abruptly, rather make a grand conclusion.

24. Line 18. “The goal is to achieve herd immunity…” The goal of vaccination is not mere achieving herd immunity. Please reconsider it.

25. Lines 310-320. Please use initials only instead of full names.

26. Lines 335-337. If you do not wish to add Acknowledgement, please remove the whole section.

Major comments

1. As you may have already guessed from some of my comments, my main remark is concerned with your having too many social-related statements that are not necessary for your current research. They only hinder reading and understanding your results. If you rewrite lines 51-93 (three paragraphs of Introduction) so that excessive political and social theorizing might be avoided, that would be commendable. Please condense these three paragraphs in three-four sentences about the situation with vaccination motivation / status for immigrants / ethnic minorities. There should be no idle theorizing that leads you to nowhere afterwards.

2. You are beginning your Abstract with “Vaccination is considered the best strategy to prevent severe illness or death from COVID-17 19 infection.” Strictly speaking, this is incorrect after two and a half years of the pandemic. Vaccinated people get ill and die too. There is different and complex situation in different age groups, comorbidities, and – more importantly – for different immunity particularities. So, please reconsider your first sentence in the Abstract or simply remove it.

3. Several times you stressed that in the immigrant Hispanic/Latine communities the audience was substantially older (the statistically important difference) than in the US-born group (e.g., lines 174, 242). Why? Please explain it very carefully and in detail in either Results or Discussion. Was it a mistake of planning / performing the survey? A coincidence? A rule? Or did you designed your research in such a way? Then what did you pursue by that difference of age?

4. You major result is that despite SARS-CoV-2 vaccination propensity / current status, after making allowance for age and race, being not significantly different between U.S.-born and foreign-born participants, the immigrant community demonstrates a much greater motivation for vaccination (e.g., lines 304-306). Please add your ideas why this may be so, to Discussion. Discussing your main result will help a reader to understand it better.

Thank you.

Yours respectfully,
The Reviewer

Author Response

Thank you so much.

Reviewer 2 Report

In the “introduction” there is an excellent analysis of the socio-cultural health determinants in native Hispanics in the US, compared to immigrant Hispanics, and the so-called Hispanic epidemiological paradox is well explained.

In the “methods” are cited two studies from 2021 from which the questionnaire used in this study was drawn: the 2 studies should be reported with notes and cited in the references.

To two portions of sample are cited, one of 83 subjects and one of 44: since the sample size appears small (as correctly underlined in the weaknesses in the "discussion" section of the study), it would be good to underline the characteristics of these people (e.g. are they opinion leaders capable of acting on the behaviour of the individuals of their population?)

In any case, all numbers must be moved to the "results" section. In the methods, only the characteristics of the sample recruited must be described.

It is correct to start the "results" section with the number of subjects in the sample and its composition, or to quote table 1, where these elements are reported.

Lines 185 and 186: enter only the results of the binary logistic regression and not the reasons why this analysis was performed (to be explained in the methods instead).

In the “results” is highlighted the role of the so-called "familism" as a conditioning factor in adherence to vaccination, despite a lower cultural level in Hispanics born outside the US than in the USA.

Also correct is the analysis of the differences between acculturation and enculturation and of the integrations of these processes for public health purposes, through adherence to vaccinations.

The sample size is recognized as a limit and it would be good to better explain why the sample is so limited.

Finally, it would be interesting to read concrete proposals for Public Health actions in collaboration with the aforementioned associations (CEAL, NIMHD, NHLBI), to consolidate this approach, not only towards the booster doses of COVID-19 vaccine which have just been approved, but also towards all other vaccines (see emergency Poliomyelitis).

Finally, in support of this detailed analysis, the vaccination coverage data in the two populations under study (Hispanics native in the US and Hispanics not native) could be cited.

Author Response

Thank you so much.

Round 2

Reviewer 1 Report

Dear authors,

Since you have made extensive work on your text and most of my remarks have been given due attention, I recommend your article for publishing. Thank you.

Reviewer